# Effect of PGPR on growth and nutrient utilization of *Elymus nutans* Griseb at different temperatures

Linling Ran[1⊙], Haoyang Wu[1], Fei Xia[2,3⊙], Yunyin Xue[4], Wei Wei[2,3], Junqiang Wang🄳[1*], Jinglong Wang[2,3], Shanshan Zhao[1], Shuang Yan[1], Hao Shi[5], Shaikun Zheng[5], Yu Zhang[1], Xiaoqin Qiu[1]

**1** Key Laboratory of Southwest China Wildlife Resource Conservation (Ministry of Education), China West Normal University, Nanchong, China, **2** State Key Laboratory of Highland Barley and Yak Germplasm Resources and Genetic Improvement, Lhasa, China, **3** Institute of Pratacultural Science, Tibet Academy of Agriculture and Animal Husbandry Science, Lhasa, China, **4** School of Geography and Tourism, Shaanxi Normal University, Xi'an, China, **5** College of Forestry, Gansu Agricultural University, Lanzhou, China

⊙ These authors contributed equally to this work and should be considered co-first authors.
* wangjunq0303@163.com

## Abstract

Plant growth-promoting rhizobacteria (PGPR) are beneficial bacteria that facilitate plant growth and can be used in the restoration of ecosystems. However, PGPR vary in their temperature tolerance, and few studies have investigated the effect of temperature on PGPR-mediated growth promotion or PGPR inoculum colonization. Therefore, we isolated and purified rhizosphere bacteria from the rhizosphere soil of *Elymus nutans* Griseb (*En*G), collected from the Qinghai-Tibet Plateau. Selective culture media were used to assess whether these strains possess plant growth-promoting abilities and to measure the magnitude of their plant growth-promoting ability. Then screen out the strains (S1, S2, S3, S4, and S5) with strong plant growth-promoting ability for identification. To demonstrate the growth-promoting effects of the selected PGPR, we conducted a study. In this study, we simulated three temperature gradients (10°C, 15°C, and 20°C) during the growing season of *En*G on the Tibetan Plateau. Furthermore, we established four incubation substrate treatments: T1(addition of PGPR but no addition of NPK fertilizers), T2 (neither PGPR nor NPK fertilizers addition), T3 (addition of PGPR both and NPK fertilizers), and T4 (addition of NPK fertilizers but not PGPR), to explore the effects of PGPR on the growth and nutrient (NPK) utilization efficiency of *En*G at different temperatures. The results revealed that compared with those under T2, the plant height (PT) and dry weight under, T1 increased by 51.72% – 70.67% and 24.99–51.25%, respectively. The soluble sugar (SS) and soluble protein (SP) content significantly increased by 59.37% and 369.66%, respctively, at 10 °C ($p < 0.05$) and by 100.17% and 94.5%, respectively, at 15 °C ($p < 0.05$). Compared with those under T4, the physiological efficiencies of N (NPE) at 15 °C and 20 °C significantly decreased by 40.43% and 72.11%, respectively, under

**Data availability statement:** Data used in this study can be found in Figshare (https://doi.org/10.6084/m9.figshare.27094078.v1).

**Funding:** This work was financially supported by the Science and Technology Program of Tibet Autonomous Region (XZ202201ZY0005N), Sichuan Natural Science Foundation Project (2024NSFSC2074), National Natural Sciences Foundation of China (NO.41867013), China Forestry and Grassland Reform and Development Fund (GZFCG2023-17620), and Innovation Team Funds of China West Normal University (KCXTD2023-5). We also thank the anonymous reviewers for providing critical comments and suggestions that improved the manuscript. The funder actively participated in the study design, data gathering and analytical processes, as well as in making the decision regarding publication and manuscript compilation.

**Competing interests:** The authors declare that the research was conducted in the absence of any commercial or financial relationships that could be construed as a potential conflict of interest.

T3. In summary, these showed that this PGPR (S1, S2, S3, S4, and S5) promoted the growth of *En*G on the Tibetan plateau and improved its nutrient utilization efficiency.

## 1. Introduction

Under natural conditions, plants continuously interact with a microbial community known as the phytomicrobiome [1]. Certain soil microorganisms enhance the enhancement of plant growth and development [2]. Plant growth-promoting rhizobacteria (PGPR) are beneficial bacteria that colonize plant roots and enhance plant development [3]. PGPR include numerous soil bacteria species that promote the growth of host plants that are cultivated in their presence [4]. These microbes populate the root zone of plants and boost plant growth either directly or indirectly [5]. Wu [6] showed that the application of *Bacillus megaterium* and *Bacillus mucilaginous* as microbial inoculants enhanced plant growth and improves the plant's uptake of nutrients (total nitrogen, N; phosphorus, P; and potassium, K). These microbes are essential for the solubilization of P and K and as well as N fixation. They transform insoluble P and K in the soil into plant-accessible forms via acidolysis, chelation, and exchange reactions, thereby promoting plant growth [7]. In addition, PGPR can regulate phytohormone levels in plants and thereby influence plant growth [8].

Plants face diverse environmental challenges that substantially hinder their growth and development, potentially causing substantial crop losses in agriculture. Cold temperatures represent a key abiotic factor that restricts the growth and yield of plants [9]. Extensive evidence suggests that suitable management techniques, including the use of fertilizers and external protective compounds, can mitigate the adverse effects of cold stress on plant growth [10]. PGPR can enhance plant nutrient uptake and availability through biochemical and physiological processes [11]. They also improve plant stress tolerance by inducing systemic resistance, producing phytohormones, and solubilizing nutrients [12,13]. Nevertheless, different PGPR strains exhibit considerable differences in their temperature tolerance, and these differences directly affect their ability to promote plant growth [14]. Therefore, selecting PGPR strains that are suitable for a particular forage and its environment is essential.

*Elymus nutans* Griseb (*En*G) is a native and widespread grazing grass commonly found in the highland pastures of the Qinghai-Tibetan Plateau in China [15]. As an excellent pasture grass, it has strong cold resistance and tolerance, rich nutrients, good palatability, high yield, and drought tolerance. It can also prevent winds and fix sands, conserve soil and water, and accelerate the process of ecological restoration of the grassland, which can effectively reduce the economic losses caused by natural disasters [16]. Therefore, it is widely planted to restore degraded alpine meadows on the Qinghai-Tibet Plateau [17].

The growth of *En*G relies on symbiotic relationships between its roots and microbes, which are essential for nutrient uptake and enhancing growth in its habitat [18]. Therefore, investigating the PGPR in the rhizosphere of *En*G on the Qinghai-Tibetan Plateau is crucial. Several studies have demonstrated that PGPR can fix nitrogen, solubilize phosphorus, enhance potassium availability, and secrete

indole-3-acetic acid (IAA), thereby promoting plant growth [19–22]. However, few studies have examined the effects of temperature on PGPR-mediated growth promotion or PGPR inoculum colonization [23]. To systematically elucidate the growth-promoting mechanism of PGPR at different temperatures and to improve the yield of *En*G, we aimed to: (1) investigate the effects of PGPR on the growth and nutrient utilization of *En*G, and (2) evaluate the effects of PGPR on *En*G under varying temperature conditions. The findings of this study would help identify valuable microbial resources and provide foundational insights, thus supporting the development and application of PGPR as a biofertilizer to enhance pasture productivity on the Tibetan Plateau.

## 2. Materials and methods

### 2.1. Sources of PGPR

The sampling site a is located in Nagqhu Town, Seni District, Nagqhu City, Tibet Autonomous Region, with geographic coordinates of 31°26′N latitude and 92°07′E longitude (Fig 1), and an average elevation of 4450 m. Three sampling sites with good growth and disease-free *En*G were randomly selected, and the rhizosphere soil of the *En*G was collected using root shaking method, with five replicate mixes randomly sampled at each site. The rhizosphere soil, together with the fine roots, were put into sterile self-sealing bags and stored at low temperature (4°C) to be brought back to the laboratory. The collected soil samples were prepared into $10^{-4}$ and $10^{-5}$ soil suspensions and 100 µL of soil suspension was aspirated and spread evenly on beef paste peptone agar plates for bacterial purification and isolation of bacteria.

### 2.2. Determination of the growth-promoting ability of PGPR and screening of high-quality bacterial strains

Strains were inoculated onto nitrogen-free medium (NFM), Pikovskaya medium, Bacterial Organophosphorus medium, Potassium Bacteria medium, and King's B medium to assess their growth-promoting abilities. Among them, 273 strains exhibited growth on the selective medium. The ability of the isolates to grow on NFM agar and their nitrogen-fixing capacity was evaluated using the acetylene reduction method [24]. Strains' phosphorus solubilizing capacity was tested with Mo blue colorimetry [25]. The strains were cultured in 50 mL King's broth at 28 °C with shaking (180 rpm) for 3 days. IAA was extracted from the culture filtrate using ethyl acetate, with each extraction performed thrice [26]. The IAA content was measured using high-performance liquid chromatography [27]. The KSB-containing solution was diluted to 50 mL and centrifuged at 500 rpm for 10 minutes to eliminate solids. Subsequently, a 10 mL aliquot was further centrifuged at 10,000 rpm for 10 minutes, and the K in the supernatant was quantified using flame spectrophotometry [28]. Five strains (S1, S2, S3, S4, and S5) with high growth-promoting ability were identified through growth-promoting assays. All the five strains solubilized inorganic and organic phosphates, decomposed K, and secreted IAA. In addition, S1, S3, S4, and S5 fixed N (Table 1).

### 2.3. The 16S rRNA gene was amplified by polymerase chain reaction (PCR)

The five selected strains were inoculated in LB liquid medium for incubation at 37 °C under shaking (160 rpm) for 24 h. Bacterial genomic DNA was extracted from these strains using a commercial kit (Tiangen Biochemical Technology, Beijing, China). The 16S rRNA using polymerase chain reaction (PCR) with the primer set 27 F (5′-AGAGTTTGATCCTGGCTCAG-3′) and 1492R (5′-GGTTACCTTGTTACGACTT-3′) as described by Galkiewicz and Kellogg (2008) [29]. The PCR was performed in a 50 µL volume, comprising 1 µL of 5 µM forward primer, 1 µL of 5 µM reverse primer, 25 µL of 2 × Taq PCR MasterMix, 2 µL of DNA template, and double-distilled water. The PCR thermal cycling parameters were as follows: initial denaturation at 95 °C for 5 minutes, followed by 30 cycles of 95 °C for 30 seconds, 57 °C for 60 seconds, and 72 °C for 30 seconds, with a concluding extension phase at 72 °C for 10 minutes. The PCR results were validated through 1% agarose gel electrophoresis and subsequently subjected to sequencing via the dideoxy chain-termination approach at Sangon Biotech Co., Ltd. (Shanghai, China) [30]. The sequences of the 16S rRNA genes from the five bacterial strains have been submitted and archived in the GenBank repository, and corresponding accession numbers have been assigned— PQ351188, PQ351189, PQ351190, PQ351191, and PQ351192.

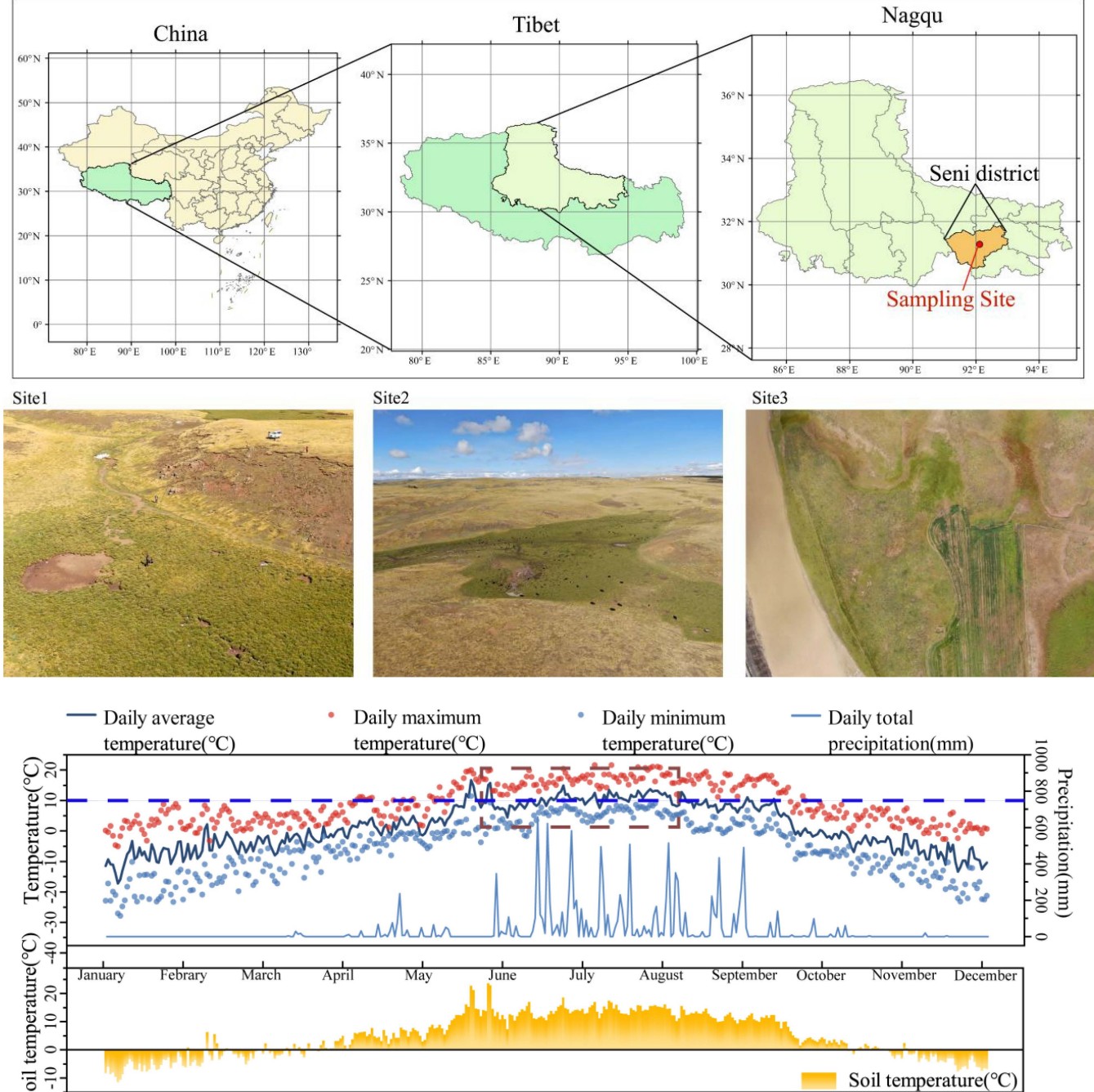

**Fig 1. General location and climatic parameters of three sampling sites.** Three randomly selected areas where *Elymus nutans Griseb* was growing vigorously. The map, base map, shape files, and map data were obtained from the National Geographic Information Public Service Platform "TianDiTu" (https://cloudcenter.tianditu.gov.cn/dataSource), the specific URL of the web page for retrieving this data is: https://cloudcenter.tianditu.gov.cn/administrativeDivision/. The map base layer has been reviewed and approved with the approval number GS (2024) 0650. The base map is unmodified.

**Table 1. Growth promoting characteristics of five PGPR strains.**

| Strains | Nitrogen fixation capacity (µg mL$^{-1}$) | Inorganic phosphorus solubilization capacity (µg mL$^{-1}$) | Organic phosphorus solubilization capacity (µg mL$^{-1}$) | Potassium-solubilising capacity (µg mL$^{-1}$) | IAA contents (µg mL$^{-1}$) | Genus | Strain number |
|---------|-----------|-----------|-----------|-----------|-----------|-------|---------|
| S1 | 5.10±0.07 | 176.2±62.46 | 24.73±15.91 | 4.42±0.34 | 596.87±63.7 | *Serratia* | PQ351188 |
| S2 | – | 136.32±68.19 | 79.85±5.34 | 11.42±0.59 | 197.10±22.80 | *Pseudomona* | PQ351189 |
| S3 | 2.08±0.06 | 250.24±30.11 | 44.52±26.51 | 8.18±0.42 | 193.32±7.17 | *Bacillus* | PQ351190 |
| S4 | 5.06±0.02 | 172.09±31.22 | 47.34±4.24 | 6.05±0.35 | 168.77±21.35 | *Acinetobacter* | PQ351191 |
| S5 | 5.57±0.03 | 311.95±50.06 | 49.46±5.61 | 4.59±0.44 | 166.64±16.88 | *Bacillus* | PQ351192 |

### 2.4. Mixed bacterial solution and incubation substrate preparation

The compatibility between selected strains (S1-S5) was verified using the plate confrontation method [31]. The selected strains (S1-S5) were inoculated into sterilized LB liquid medium respectively, and activated and cultured in a constant temperature shaker at 30 °C and 150 r/min for 12 h. The $OD_{600}$ value of each activated bacterial solution was adjusted to 0.6. Samples were then transferred to fresh LB medium, and incubated in a constant temperature shaker at 30°C and 150 r/min for 16 h to expand the culture. After centrifugation, the supernatant was discarded, and the bacterial pellet was resuspended in sterile water, fully mixed, and prepared as a single bacterial agent. After sufficient mixing and make the single bacterial agent with $OD_{600}$ value of 0.6 (the concentration of the single bacterial agent is about $1 \times 10^8$ CFU/ml). Equal amount of mixing to prepare a composite bacterial agent [32].

The culture substrate was prepared according the soil characteristics of the sampling sites. It was prepared by blending peat soil, vermiculite, perlite, and coir in a ratio of 4:1:1:1 (v/v). Each bag contained 50 g per bag of the substrate, which was then sterilized at 121 °C for 30 minutes. Subsequently, 78.2 g of the sterilized substrate was mixed with 1% (volume/volume) PGPR bacterial mixture and transferred into a 660 mL plastic container (70 × 95 × 98 mm) [33]. A control was prepared by placing an equal volume of sterile water in a cup.

### 2.5. Incubation experiment

This study included the following four treatments, each with three replicates: T1(addition of PGPR and no addition of NPK fertilizers), T2 (neither PGPR nor NPK fertilizers), T3 (addition of PGPR and NPK fertilizers), and T4 (no addition of PGPR, but addition of NPK fertilizers). The NPK were applied at rates of 150 mg kg$^{-1}$, 120 mg kg$^{-1}$, and 90 mg kg$^{-1}$ of urea (N,46.65%), superphosphate ($P_2O_5$,12%) and potassium chloride ($K_2O$,60%), respectively. Furthermore, based on the climatic temperature during the growing season (June, July, and August) of *En*G on the Tibetan Plateau (Fig 1), three incubators with temperature gradients of 10 °C, 15 °C, and 20 °C. The incubation was conducted in three incubators maintained at a light (16 h)/dark (8 h) cycle (light at 350 µEm$^{-2}$ s$^{-1}$) and 60% humidity. Plant seeds (wild *En*G seeds collected in Nagqhu, Tibet) were sterilized on the surface with a 1:1 mixture of 30% hydrogen peroxide and 70% ethanol for 10 minutes and cleaned repeatedly with demineralized sterile water. Thirty-five *En*G seeds were planted in each cup. In order to better simulate the plant growth cycle in alpine meadows, plant growth indicators were measured after 120 days of plant growth (based on the local plant growth season).

### 2.6. Indicators measurement

#### 2.6.1. Dry weight (DW), plant height (PT), and root system assay. Fifteen plants per pot were randomly sampled to determine plant height (PT). After drying at 65 °C for 24 h, the dry weight (DW) was noted. Root length (RL), the number of

root branches (RB), root surface area (RSA), and the average root diameter (RD) were all assessed using an LA2400 root scanner (Jeili Electronics Trading Co., Ltd, Shanghai, China) [34].

**2.6.2. Plant chlorophyll (Chl), soluble sugars (SS), and soluble proteins (SP) content assay.** To measure the chlorophyll (Chl) content, 0.1 g of fresh leaf tissue from each seedling was mixed with a bit of quartz sand, ground in 80% acetone, and then centrifuged at $9000 \times g$ and 4 °C for 10 minutes. The supernatant's absorbance was read at 440, 663, and 645 nm to assess Chl a, or 10 min. The absorbance of the supernatant was read at 440, 663, and 645 nm to assess Chl a, Chl b, and carotenoid levels, respectively. The total Chl content was calculated using the method described by Li et al. [18]. The soluble sugar (SS) content was assessed following the method ofoutlined by Gurrieri et al. [35]. Soluble protein (SP) content was determined using Bradford's method for staining with Caulmer Brilliant Blue [36].

**2.6.3. Plant nutrient use efficiency.** The physiological efficiencies (PEs) of N (NPE), P (PPE), and K (KPE) were calculated as the kilogram of plant yield per kilogram of N, P, and K uptake, respectively [37]. The agronomic efficiencies (AEs) of N (NAE), P (PAE), and K (KAE) were calculated as the yield increase per kilogram of N, P, and K applied, respectively [38,39]. The apparent recovery efficiencies (REs) of N (NRE), P (PRE), and K (KRE) were calculated as the plant N, P, and K uptake (kg ha$^{-1}$) per kg of N, P, and K applied, respectively [40].

## 2.7. Statistical analyses

Statistical analyses were performed using the SPSS software (version 27.0; SPSS Inc., Chicago, IL, USA). In the context of our research, prior to conducting parametric tests on each dataset, we initiated a preliminary assessment using the Shapiro-Wilk test to ascertain the conformity of the data to a normal distribution pattern. Mean differences were determined using the the Tukey test at a significance level of 5%. Statistical significance is reported at $p < 0.05$. All visualizations were produced using R 4.4.1. The relationship between the growth indicators of EnG and nutrient utilization was analyzed through Mantel analysis.

## 3. Result

### 3.1. Effects of PGPR on EnG growth characteristics at different temperatures

The substrates with PGPR microbial inoculants influenced the growth characteristics of EnG (Fig 2). Compared to neither PGPR nor NPK fertilizers treatment, the addition of PGPR but no addition of NPK fertilizers treatment increased PT by 51.72% (10 °C) to 70.67% (20 °C), and increased DW by 24.99% (15 °C) to 51.25% (10 °C) (Fig 3). Compared to no addition of PGPR but addition of NPK fertilizers treatment, addition of PGPR and NPK fertilizers treatment significantly increased PT by 30.66% (20 °C) to 55.22% (10 °C), and significantly increased DW by 4.86% (10 °C) to 94.71% (20 °C) ($p < 0.05$) (Fig 3a and 3c). Treatment T1 and T3 with the addition of PGPR bacterial liquid significantly increased the Chl of EnG compared to T2 and T4 without the addition of PGPR bacterial liquid ($p < 0.05$) (Fig 3c).

### 3.2. Effects of PGPR on EnG root growth at different temperatures

The effects of PGPR on the roots of EnG are presented in Fig 4. Compared to neither PGPR nor NPK fertilizers treatment, the addition of PGPR but no addition of NPK fertilizers treatment significantly increased RSA of EnG by 96.32% and 109.31% at 10 °C and 15 °C, respectively ($p < 0.05$) (Fig 4). Compared with the no addition of PGPR but addition of NPK fertilizers treatment, the addition of PGPR and NPK fertilizers treatment increased the RL, RB, RSA, and RD of EnG. Furthermore, the RL of EnG was significantly higher at 15 °C than at 20 °C ($p < 0.05$) (Fig 4a).

### 3.3. Effects of PGPR on the SS and SP contents of EnG at different temperatures

Compared to neither PGPR nor NPK fertilizers treatment, the addition of PGPR but no addition of NPK fertilizers treatment significantly increased the SS and SP of EnG by 59.37% and 369.66% at 10 °C, by 100.17% and 94.5% at 15 °C, respectively ($p < 0.05$) (Fig 5). Furthermore, the content of SS at 10 °C under the addition of PGPR and NPK fertilizers

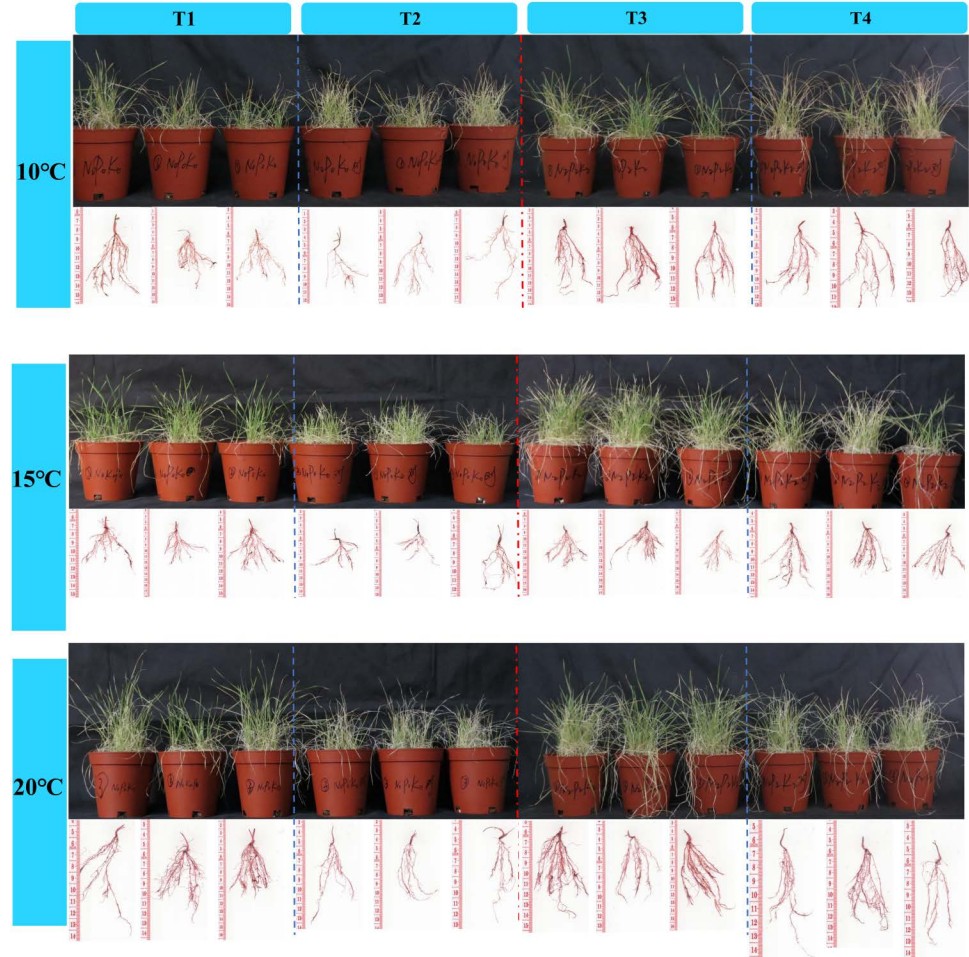

**Fig 2. The effects of different treatments on the growth of *En*G at different temperatures.** T1(addition of PGPR but no addition of NPK fertilizers), T2 (neither PGPR nor NPK fertilizers), T3 (addition of PGPR and NPK fertilizers), and T4 (no addition of PGPR but addition of NPK fertilizers).

treatment was significantly higher than that of SS at 20 °C (Fig 5a). The contents of SS and SP at 10 °C under addition of PGPR but no addition of NPK fertilizers treatment were significantly higher than at 20 °C ($p < 0.05$) (Fig 5).

### 3.4. *Effects of PGPR on En*G *nutrients use efficiency indexes at different temperatures*

Compared with no addition of PGPR but addition of NPK fertilizers treatment, the addition of PGPR and NPK fertilizers treatment significantly decreased NPE of *En*G at 15°C and 20°C by 40.43% and 72.11%, respectively ($p < 0.001$) (Fig 6a). However, compared with no addition of PGPR but addition of NPK fertilizers treatment, the addition of PGPR and NPK fertilizers treatment significantly decreased the PRE of *En*G ($p < 0.001$) (Fig 6h). Furthermore, the addition of PGPR and NPK fertilizers treatment significantly increased the NAE, PAE, KAE, NRE, and KRE of *En*G at 15°C and 20°C ($p < 0.001$) (Fig 6).

### 3.5. Relationship between PGPR and the growth indicators of EnG

The PGPR significantly affected PT, DW, Chl, RL, RSA, RD, SS, and SP (Table 2). The interaction between temperature and PGPR significantly affected the RL of *En*G (Table 2) ($p < 0.001$). In addition, RL was significantly and positively correlated with AE (Fig 7b) ($p < 0.05$). PT and DW were significantly correlated with NAE, PAE, and KAE (Fig 7a) ($p < 0.001$),

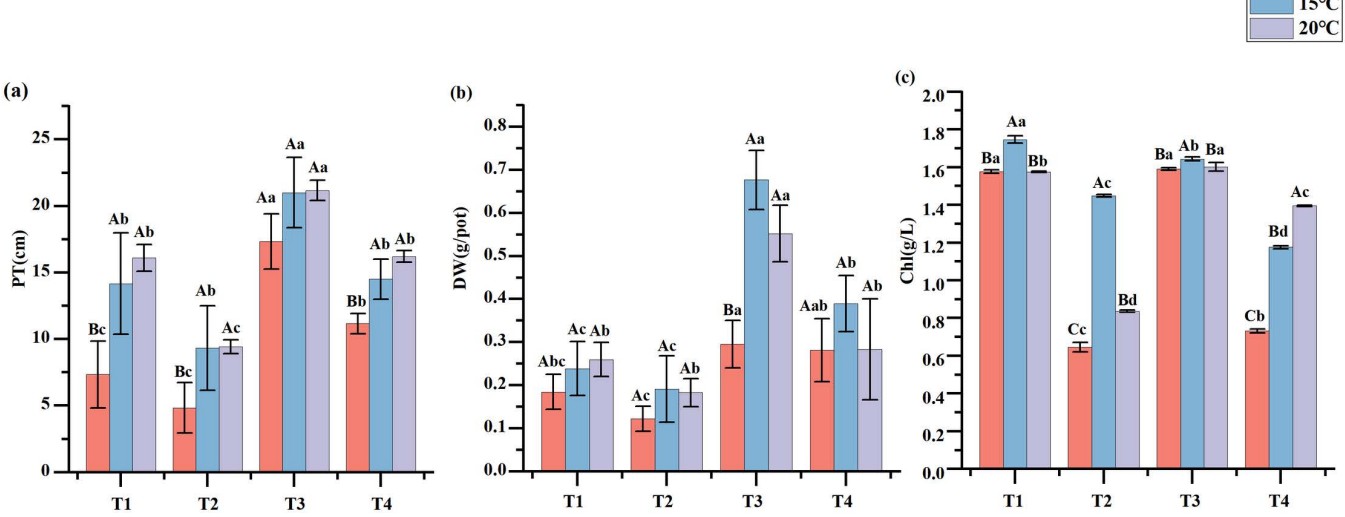

**Fig 3. Effect of different treatments on plant height (PT), dry weight (DW), and plant chlorophyll (Chl)content.** Lowercase letters indicate differences between treatments and uppercase letters indicate differences between temperatures.

and PT and DW were significantly correlated with the NRE, PRE, and KRE (Fig 7a) ($p < 0.05$). In addition, RL was significantly and positively correlated with AE (Fig 7) ($p < 0.05$).

## 4. Discussion

### 4.1. *The effects of PGPR on EnG growth*

The plant microbiome is a key plant characteristic influencing crop yield. Olasupo et al. [41] applied various *Bacillus* strains to peppers, which resulted in a 27%–36% increase in biomass. Similarly, Aw et al. [42] isolated three As-resistant PGPR strains to assess their roles in promoting rice growth. In the present study, compared to the treatment without adding either PGPR or NPK fertilizers, the addition of PGPR but no addition of NPK fertilizers treatment increased the PT and DW of *En*G (Fig 3). These findings corroborated the efficacy of PGPR in enhancing the growth of *En*G. Furthermore, the addition of PGPR and NPK fertilizers treatment significantly increased the PT and DW of *En*G compared with no addition of PGPR but addition of NPK fertilizers treatment ($p < 0.05$) (Fig 3a and 3c). There was also a significant correlation between PT, DW, AE, and RE (Fig 7a) ($p < 0.05$). This correlation underscores the role of the culture substrate with PGPR in facilitating *En*G nutrient uptake. These results indicate that application of PGPR can effectively improve soil nutrient use effectively and reduce reliance on chemical fertilizers and mitigate the negative effects of excessive fertilization [40]; accordingly, *En*G biomass can increase, and economic and ecological development can be achieved.

Plant growth requires the accumulation of photochemical products and is directly linked to the photosynthetic rate [41]. del Rosario Cappellari [43] evaluated the growth characteristics and Chl levels in peppermint (*Mentha piperita*) seedlings exposed to PGPR strains. They found that the treated plants showed significantly better growth and higher Chl levels than the untreated ones. Consistent with these findings, our addition of PGPR, but no addition of NPK fertilizers, significantly increased the chlorophyll content of *En*G compared with the treatment of neither PGPR nor NPK fertilizers ($p < 0.05$) (Fig 3c). This indicated that the culture substrate with PGPR positively influenced on the photosynthesis of *En*G. Notably, at 10 °C and 15 °C, the Chl contents under T1 and T3 were similar. Under T1, although nitrogen, phosphorus and

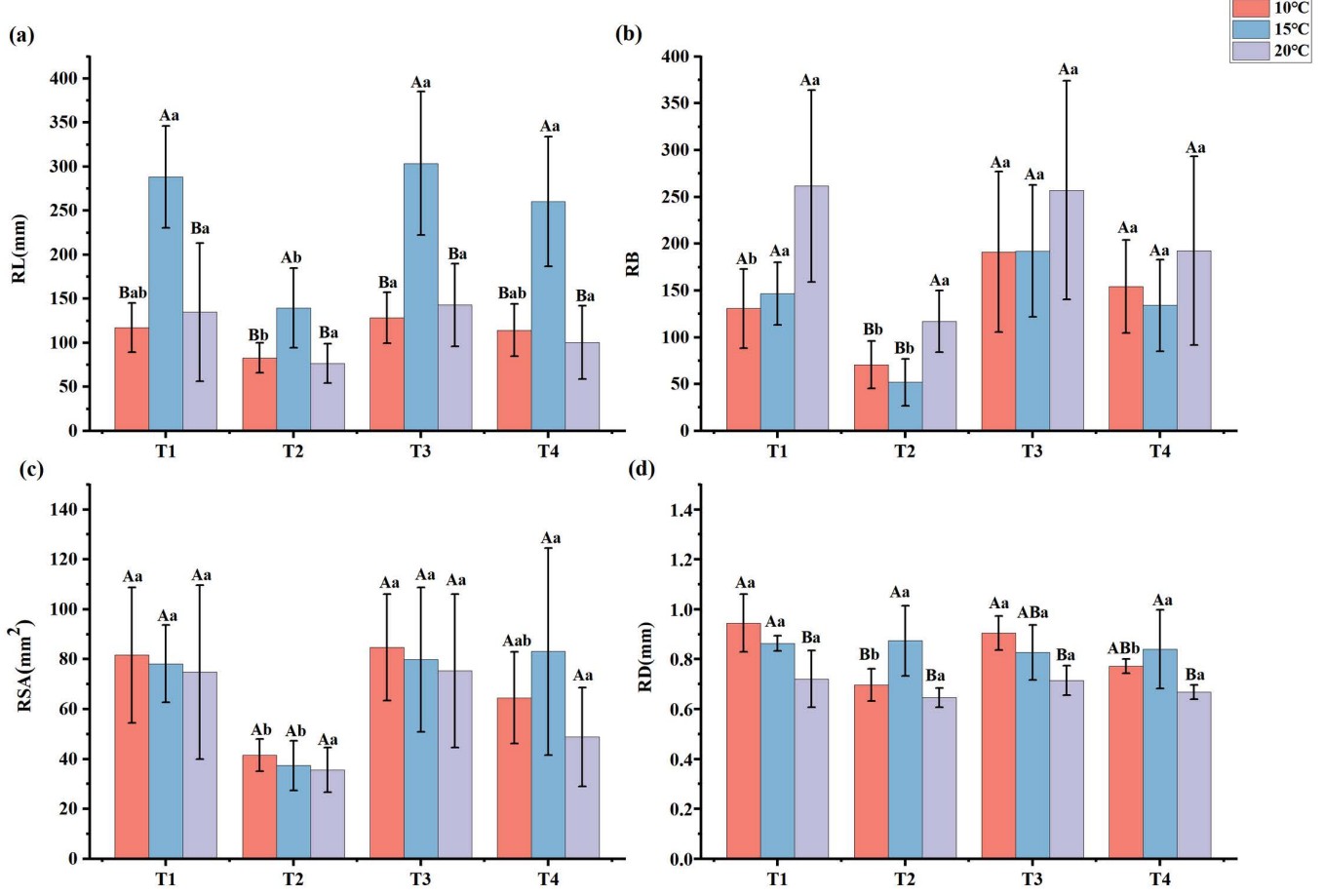

**Fig 4. Effect of different treatments on root length (RL), number of root branches (RB), root surface area (RSA), and root average diameter (RD).** Lowercase letters indicate differences between treatments and uppercase letters indicate differences between temperatures.

potassium fertilizers were not added, PGPR might have decomposed the insoluble nutrients in the culture medium through its growth-promoting characteristics, enabling plants to utilize the nutrients in the culture medium, more effectively and thus maintain a relatively high chlorophyll content [44].

Furthermore, we found that treatment addition of PGPR but no addition of NPK fertilizers treatment significantly increased the RSA of *En*G at 15 °C by 109.31% compared with neither PGPR nor NPK fertilizers treatment ($p < 0.05$) (Fig 4). This may be because the screened PGPR strains secreted IAA (Table 1). A previous study revealed that the impact of PGPR on root growth is connected to the synthesis of IAA, a substance known to advance root expansion and enhancement, augment the area of the root surface, and stimulate root metabolic processes [19]. The expansion of the root system surface area augments the nutrient absorption area, which is beneficial for facilitating the uptake of nutrients and water [45]. Notably, a significant correlation was observed between RSA levels and PE (Fig 7) ($p < 0.01$). This indicated that the PGPR is capable of enhanced the nutrient uptake efficiency of *En*G by promoting root growth [43].

### 4.2. The effect of PGPR on the nutrient utilization of EnG

Nitrogen (N), phosphorus (P), and potassium (K) are fundamental chemical elements essential for plant development, contributing significantly to plant tissue formation and physiological processes [46]. The effect of PGPR on plant nutrition

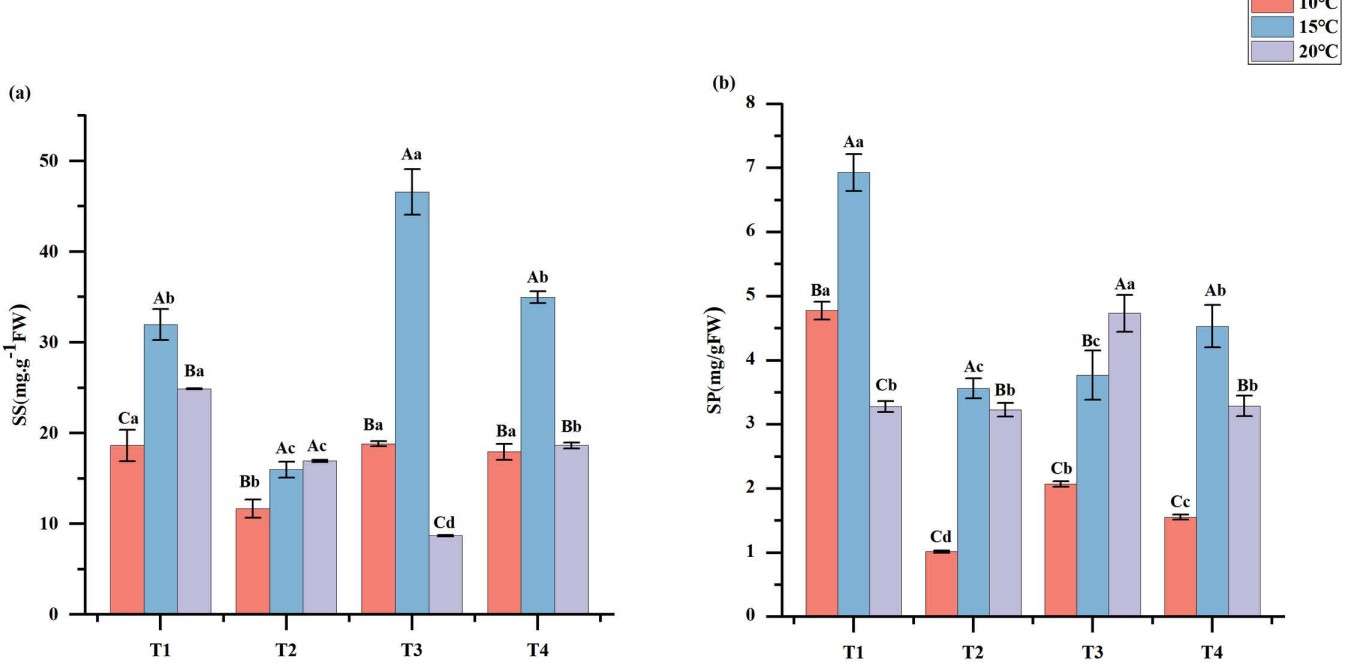

**Fig 5. Effect of different treatments on soluble sugars (SS), and soluble proteins (SP)content.** Lowercase letters indicate differences between treatments and uppercase letters indicate differences between temperatures.

are attributed to their impact on nutrient absorption by plants [47]. Wu et al. [6] demonstrated that *Bacillus megaterium* and *Bacillus mucilaginous* inoculants not only increased plant growth but also enhanced the nutrient-absorption capabilities of plants, particularly in terms of total N, P, and K. In this study, the addition of PGPR and NPK fertilizers treatment significantly increased the NAE, PAE, and KAE of *En*G at 15°C and 20°C. This indicates that the PGPR is capable of enhancing the nutrient uptake efficiency of *En*G, thereby reducing the dependence on chemical fertilizers, and thereby promoting sustainable agricultural development [48]. Additionally, the plants root system performs several functions, including nutrient acquisition, water uptake, stability, and symbiotic relationships with beneficial soil microorganisms, which collectively enhance nutrient absorption efficiency [49]. In the present study, the AE and RE were significantly correlated with RL (Fig 7a). Furthermore, NAE, PAE, KAE, and NRE were significantly positively correlated with RL (Fig 7b). These findings affirm that efficient nutrient uptake also significantly relies on the capacity of the root system to permeate the soil [50].

Compared to the treatment with no addition of PGPR but addition of NPK fertilizers, the addition of PGPR and NPK fertilizers significantly decreased the NPE of *En*G at 20 °C by 72.11% ($p < 0.001$) (Fig 6a). This may be attributed to PGPR's ability to convert unavailable nutrient forms into plant-available forms [51], thereby diminishing the plants' reliance on synthetic fertilizers. PGPR plays a significant role in agriculture by facilitating the circulation of plant nutrients and reducing the reliance on chemical fertilizers [52]. These offer valuable microbial resources and foundational data for the development and application of PGPR as biofertilizers; however, compared with no addition of PGPR but addition of NPK fertilizers treatment, the addition of PGPR and NPK fertilizers treatment significantly decreased the PRE of *En*G ($p < 0.001$) (Fig 6h). This result aligned with the findings of previous research indicating that the use of PGPR can enhance phosphorus mobilization and supply to crops in P-deficient soils [53]. Therefore, we determined that incorporating PGPR into the culture substrate promotes nutrient uptake by *En*G and can help reduce the use of chemical fertilizers in agriculture.

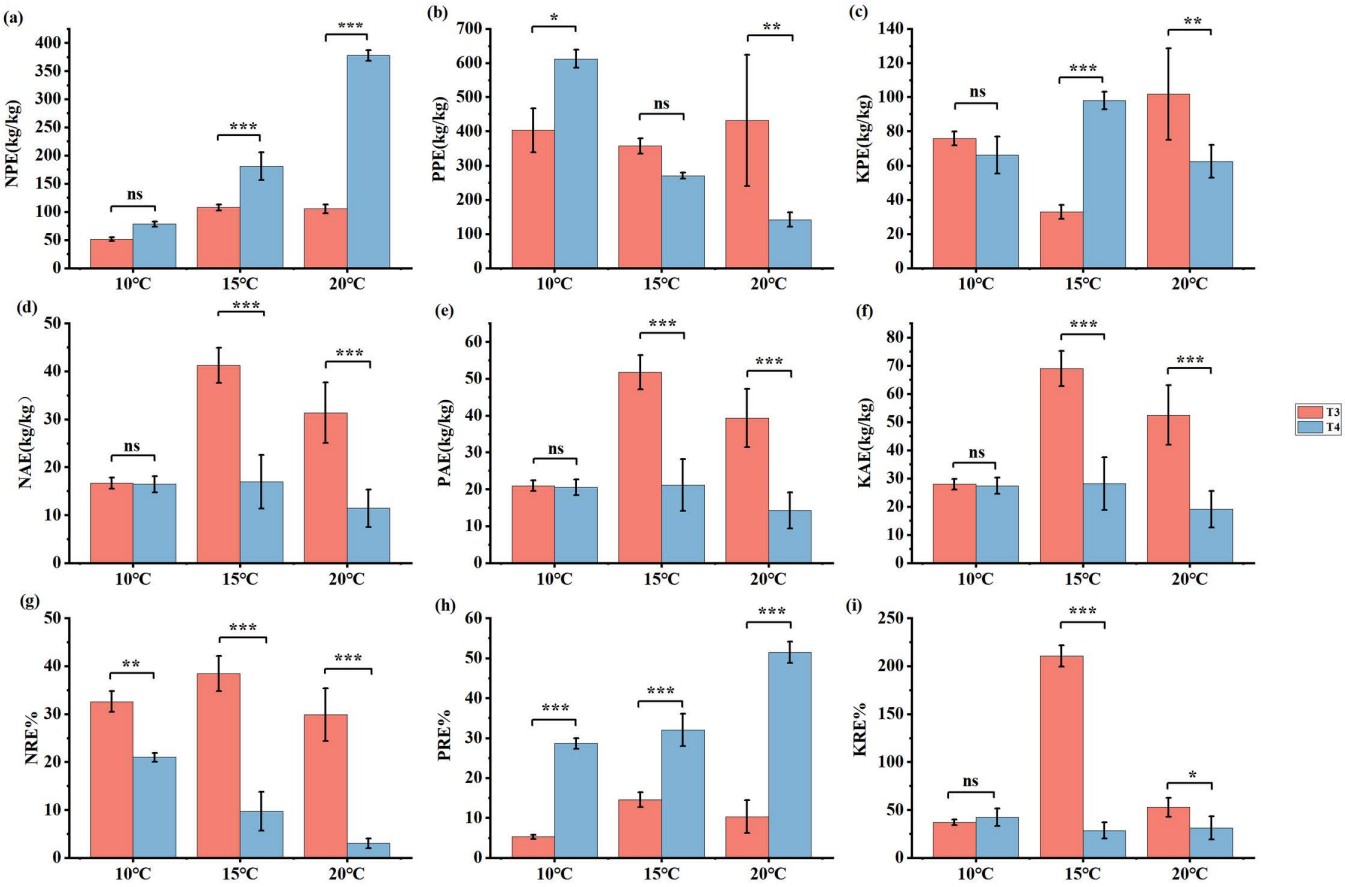

**Fig 6. Effect of different treatments (T3 and T4) on nutrient utilization.** Physiological efficiency of nitrogen (NPE), phosphorus (PPE), potassium (KPE), agronomic efficiency of nitrogen (NAE), phosphorus (PAE) and potassium (KAE), and apparent recovery efficiency of nitrogen (NRE), phosphorus (PRE) and potassium (KRE) content. * means $p < 0.05$, ** means $p < 0.01$, *** means $p < 0.001$, ns means no significant difference.

**Table 2. The main effects of temperature (T), PGPR, and fertilizers (NPK) on *Elymus nutans Griseb*.**

| Source of variation | PT | DW | Chl | RL | RB | RSA | RD | SS | SP |
|---|---|---|---|---|---|---|---|---|---|
| T | <0.001 | <0.001 | <0.001 | <0.001 | 0.076 | <0.001 | <0.001 | <0.001 | <0.001 |
| NPK | <0.001 | <0.001 | <0.001 | <0.001 | <0.001 | <0.001 | 0.847 | <0.001 | <0.001 |
| PGPR | <0.001 | <0.001 | <0.001 | <0.001 | <0.001 | 0.007 | <0.001 | <0.001 | <0.001 |
| T* NPK | 0.333 | 0.007 | <0.001 | 0.429 | 0.893 | 0.012 | 0.015 | <0.001 | <0.001 |
| T* PGPR | 0.629 | 0.028 | <0.001 | <0.001 | <0.001 | 0.392 | <0.001 | <0.001 | <0.001 |
| NPK* PGPR | 0.386 | 0.007 | <0.001 | 0.174 | 0.949 | 0.728 | <0.001 | <0.001 | <0.001 |
| T* NPK* PGPR | 0.290 | 0.026 | <0.001 | 0.020 | 0.898 | 0.010 | 0.181 | <0.001 | <0.001 |

The values in the table represent the statistical significance (*p* - values) of the effects of different sources of variation on each trait. PT: plant height; DW: dry weight, Chl: plant chlorophyll; RL: root length; RB: number of root branches; RSA: root surface area; RD: root average diameter; SS: soluble sugars; SP: soluble proteins.

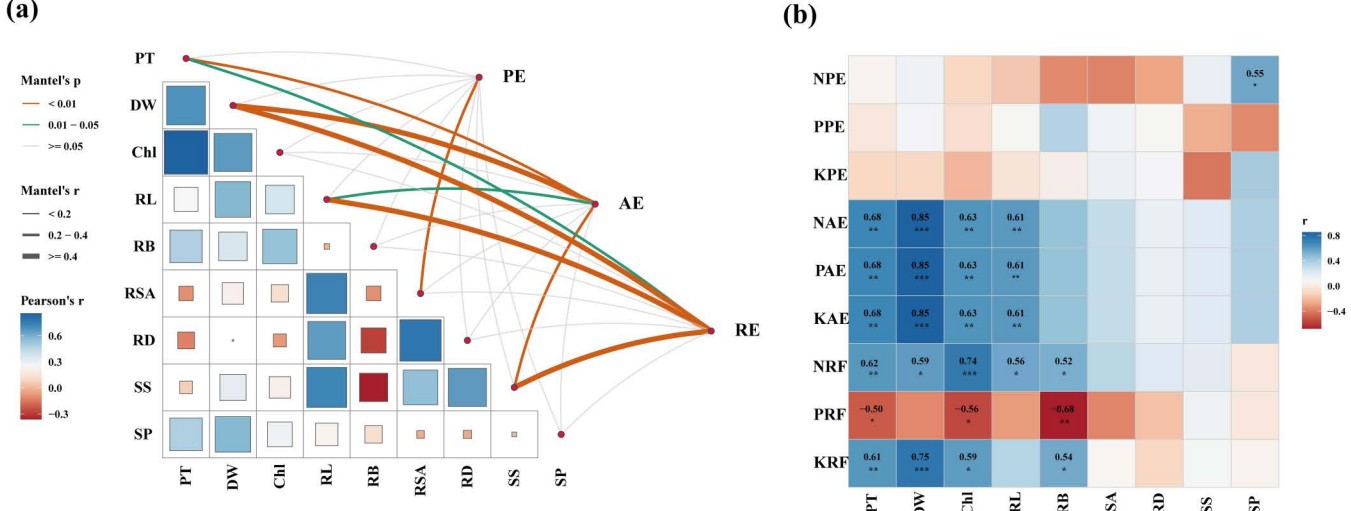

**Fig 7. Relationship between plant growth factors and nutrient utilization.** (a) Spearman's correlation between the relative abundance of PE, AE, and RE, and Plant growth factors. *, **, and *** represent significant correlations at the 0.05, 0.01, and 0.001 levels. (b) Mantel test between nutrient utilization and indicators of plant growth. In Figure (a), the thickness and color of the lines are indicated by Mantel's p/r values, illustrating the degree of correlation's significance and intensity. Figure (b) employs a heatmap, where the color gradient communicates that blue denotes a positive correlation, whereas red indicates a negative one. PT, plant height; DW, dry weight; Chl, plant chlorophyll; RL, root length; RB, number of root branches; RSA, root surface area; RD, root average diameter; SS, soluble sugars; SP, soluble proteins.

### 4.3. The effects of PGPR at different temperatures

Under low-temperature stress conditions, plants exhibit a heightened accumulation of soluble sugars (SS) such as sucrose, glucose, fructose, ribose, and trehalose [54]. These sugars are capable of neutralizing free radicals and indirectly stimulating protein synthesis, which in turn enhances cold resistance [55]. Soluble proteins (SP) are also pivotal, acting as osmoregulators and supporting low-temperature tolerance in plants [56]. Previous studies have shown that the content of SP increased up to 68% in plants treated with PGPR [57]. Additionally, PGPR-treated plants exhibited significantly higher SS content, approximately double that of the untreated controls [54]. Our findings aligned with these observations. Compared to the control group without PGPR or NPK fertilizers, the addition of PGPR but no addition of NPK fertilizers significantly increased the SS and SP contents of *En*G by 59.37% and 369.66%, respectively, at 10 °C (Fig 5). This indicated that PGPR positively influenced the physiological and biochemical parameters of *EnG* [58]. Thus, inoculating plants with PGPR can facilitate growth by enhancing their tolerance to various a types of biotic stresses [59]. Notably, under T1 and T3 treatments with PGPR inoculation, the SS and SP contents at 10 °C were significantly higher than those at 20 °C ($p < 0.05$) (Fig 5). This is likely because low temperatures enhanced the activities of enzymes involved in SS synthesis. At 10 °C, these enzymes may be more inclined to break down and convert stored complex carbohydrates, such as starch, into soluble sugars. This is because upon sensing the low - temperature environment, plants initiate a series of physiological adaptation mechanisms. Through the process of conversion, they transform less mobile nutrient forms, such as starch, into more transportable and utilizable soluble sugars. This conversion is essential for maintaining the osmotic balance and energy supply of cells [60,61].

Temperature is a key factor limiting the distribution and yield of plants [62]. It profoundly influences root growth, shapes root system architecture, and affects nutrient uptake and overall plant health [63]. In this study, the interaction between temperature and PGPR inoculation significantly affected the root length of *En*G ($p < 0.001$) (Table 2). This was attributed to the combined effects of temperature on root development and the ability of PGPR to modulate plant hormone levels and

improve nutrient and water uptake [64]. Consequently, the root length of *En*G was significantly higher at 15 °C than that at 20 °C ($p < 0.05$) (Fig 4a). However, under the influence of PGPR, the *En*G was more productive after inoculation with PGPR under temperature stress than under normal temperatures, consistent with the findings of L. B [58]. Bruno [65]. An increase in plant biomass, because of nutrient accumulation, strongly indicates that PGPR promotes plant growth [66]. Therefore, PGPR can stimulate plant growth and enhance plant resistance to unfavorable conditions by intensifying physiological activity and regulating nutrient equilibrium [65]. These findings will provide valuable resources for the restoration of forage affected by extreme weather conditions in the Qinghai-Tibetan Plateau.

## 5. Conclusions

In this study, five distinct plant growth-promoting rhizobacteria (PGPR) strains (S1, S2, S3, S4, and S5) were isolated and characterized for their growth-enhancing capabilities. Subsequent pot experiments confirmed that these PGPR strains significantly enhanced the growth of *En*G, while also improving its nutrient utilization efficiency and cold tolerance. The substrates with PGPR increased PT and DW of *En*G, as well as its SS and SP contents compared to those without PGPR. Furthermore, the substrates amended with both PGPR and fertilizer resulted in a marked increase in the apparent nutrient recovery efficiency of *En*G, which suggested that PGPR strains were efficacious in optimizing plant nutrient uptake, thereby mitigating the necessity for excessive fertilizer application. These findings provide valuable strain resources and foundational data for the formulation of specialized microbial fertilizers tailored for the alpine meadows of the Qinghai-Tibet Plateau.

## Author contributions

**Formal analysis:** Shanshan Zhao.

**Funding acquisition:** Junqiang Wang.

**Investigation:** Haoyang Wu.

**Methodology:** Yunyin Xue, Xiaoqin Qiu.

**Project administration:** Fei Xia, Wei Wei.

**Resources:** Shuang Yan, Yu Zhang.

**Software:** Yunyin Xue.

**Supervision:** Jinglong Wang, Shaikun Zheng.

**Visualization:** Yunyin Xue, Hao Shi.

**Writing – original draft:** Linling Ran.

**Writing – review & editing:** Linling Ran, Junqiang Wang.

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
