## [Decision Letter · Decision Letter 0]

17 Jan 2025

PONE-D-24-56644Mechanisms of growth promotion and resistance of PGPR to Elymus nutans Griseb at different temperaturesPLOS ONE

Dear Dr. wang,

Thank you for submitting your manuscript to PLOS ONE. After careful consideration, we feel that it has merit but does not fully meet PLOS ONE’s publication criteria as it currently stands. Therefore, we invite you to submit a revised version of the manuscript that addresses the points raised during the review process.

We look forward to receiving your revised manuscript.

Kind regards,

Massimiliano Cardinale, PhD

Academic Editor

PLOS ONE

Journal Requirements:

This work was financially supported by the Science and Technology Program of Tibet Autonomous Region (XZ202201ZY0005N), Sichuan Natural Science Foundation Project (2024NSFSC2074), National Natural Sciences Foundation of China (NO.41867013), China Forestry and Grassland Reform and Development Fund (GZFCG2023-17620), and Innovation Team Funds of China West Normal University (KCXTD2023-5). We also thank the anonymous reviewers for providing critical comments and suggestions that improved the manuscript.  

6. Please include a separate caption for each figure in your manuscript.

Additional Editor Comments :

Considering the criticisms of Reviewer 3, which I agree with, I suggest ro temove the term "Mechanisms" from the title. You evaluated the effects of PGPR application, not the mechanisms.

Reviewers' comments:

Reviewer's Responses to Questions

**Comments to the Author**

1. Is the manuscript technically sound, and do the data support the conclusions?

Reviewer #2: No

Reviewer #3: No

Reviewer #4: Yes

Reviewer #5: Yes

2. Has the statistical analysis been performed appropriately and rigorously? 

Reviewer #2: I Don't Know

Reviewer #3: Yes

Reviewer #4: Yes

Reviewer #5: Yes

3. Have the authors made all data underlying the findings in their manuscript fully available?

Reviewer #2: No

Reviewer #3: No

Reviewer #4: Yes

Reviewer #5: Yes

4. Is the manuscript presented in an intelligible fashion and written in standard English?

Reviewer #2: No

Reviewer #3: Yes

Reviewer #4: Yes

Reviewer #5: Yes

5. Review Comments to the Author

Reviewer #2: This is a valuable study as strains were obtained and characterized from the Tibetan Plateau, a unique and understudied ecosystem. The potential use of plant growth promoting bacteria in pasture production in this ecosystem is an interesting study area. However, it seems to me that the tested conditions (fertilizer scheme, substrate and temperature conditions) are not relevant to the application area, the Tibetan Plateau. Therefore, the results of this study are far less relevant and do not support the stated conclusions.

The results and discussion are very difficult to understand due to the use of many abbreviations, coded treatments and a lack of structure in the text. Therefore, I was also not able to verify the statistical analyses.

regarding availability of data:

- important information is lacking in Material and methods, e.g. NFM-agar, brief descriptions of methods, the preparation of inoculum (CFU counts etc)

- The manuscript does not disclose which five strains out of 273 strains were selected, and based on which criteria these were selected.

Reviewer #3: There is no experiment of mechanism of growth promotion and resistance of PGPR (eg. ROS, PO, PPO, PAL activity; amount of MDA, GSH, IAA, antioxidants, etc.) in this study. Here, title of this paper does not represent the results. This research is mainly focused on nutrient uptake efficiency of PGPR in peat soil based substrate with different temperature. In another words, authors should mention what and where is the PGPR mechanism in this study?

Reviewer #4: Comments for Authors

The abstract should be rewritten in a more systematic pattern to improve clarity and readability. Clarify the research aim, emphasizing the focus on temperature effects on PGPR-mediated plant growth and nutrient utilization. Specify the type of pot experiments conducted, including the temperature ranges used. Explain the significance of the treatments (T1, T2, T3, T4) briefly and consistently. Summarize the findings succinctly, focusing on the most significant results. The introduction section needs to incorporate more recent literature to provide a comprehensive background. In introduction section ([12] revealed that specific PGPR strains can impart cold resistance to rice plants, boosting their viability, fertility, and yield related traits and accelerating the progression of their reproductive cycle. Furthermore, variation in temperature tolerance exists among PGPR strains, affecting efficacy and plant health [13].), needs to rewrite. In methodology section the sub-heading “Identification of PGPR strains” is written well but its too lengthy, make it simple and short. In the result sectionIt is mention that percentage increases in plant traits like PT (plant height) and DW (dry weight). Could these increases be contextualized in terms of their biological or agricultural significance? What do these percentages mean for the practical application of PGPR in farming or plant growth?

Why were the specific temperature ranges (10 °C, 15 °C, 20 °C) chosen? Could the authors explain the relevance of these temperatures to the natural habitat or agricultural practices of Elymus nutans Griseb (EnG)?

There is a significant increase in Chl (chlorophyll content). What is the implication of increased chlorophyll content in terms of plant photosynthetic efficiency or overall health?

The root traits like RSA (root surface area) and RL (root length) are significantly affected by PGPR. Could the authors discuss how these changes in root architecture influence the overall nutrient uptake and drought resistance of EnG?

The paper mentions that RL was significantly higher at 15 °C than at 20 °C. What could be the physiological reasons behind this temperature-specific root growth pattern?

Why is there a significant increase in SS at 10 °C compared to 20 °C under the T3 treatment? Can the authors provide a rationale for this temperature-dependent response?

The increases in NAE (nitrogen agronomic efficiency), PAE (phosphorus agronomic efficiency), and KAE (potassium agronomic efficiency) are significant. Could the authors elaborate on the potential long-term benefits of these increases for sustainable agriculture?

The interaction between temperature and PGPR significantly affects RL. Could the authors discuss the possible mechanisms or environmental factors that might drive this interaction?

Discussion is written well however for the clarification of the above question modify this section accordingly.

Reviewer #5: The paper (PONE-D-24-56644) is a nice demonstration of growth promotion and resistance enhancement by PGPR to Elymus nutans Griseb at variable temperatures.

I have a few minor/major comments that authors should address before the manuscript can be accepted for publication.

Abstract is overly dense and uses unexplained abbreviations (e.g., "SS and SP") which may confuse non-expert readers.

The broader ecological and economic implications of focusing on Elymus nutans in the introduction are not sufficiently emphasized.

Put clear distinction between biological and technical replicates.

There is a lack of details on how environmental factors like light intensity and humidity were controlled across treatments.

Statistical methods (e.g., SPSS) are mentioned but without sufficient elaboration on key assumptions (e.g., normality checks).

Results:

Limited explanation for unexpected trends, such as the decrease in physiological efficiency of nitrogen (NPE) at higher temperatures in T3 treatment.

You mentioned p-values but could not discuss the biological significance of findings.

Table 2 presents complex statistical results without adequate textual interpretation for non-specialists.

For figures, legends are overly technical and could be simplified for clarity.

The color coding in heatmaps (Figure 7a) is inconsistent, making interpretation less intuitive. Also, text is not legible.

In Fig. 2, measuring scales are not legible. Better to use a common scale for all and that should be clearly visible.

Improve discussion: there are repetitive points about the benefits of PGPR, leading to redundancy.

Alternative mechanisms for PGPR effects (e.g., microbial community shifts) are not discussed.

Limited exploration of variability among the five PGPR strains tested.

Discuss limitations of the study in the conclusion, such as the exclusion of field trials or the potential for long-term ecological impacts. Give practical recommendations for scaling PGPR applications are underdeveloped.

Below are some papers for authors for scientific discussion and for probable citation:

doi.org/10.1007/s00344-023-11119

https://doi.org/10.1016/B978-0-323-91595-3.00011-2

Plant Stress 11, 100397

Plant Growth Regulation 99 (3), 449-464

https://doi.org/10.3389/fpls.2021.746780

https://doi.org/10.3390/microorganisms9122451

Environmental and Experimental Botany 200, 104911

6. PLOS authors have the option to publish the peer review history of their article (what does this mean? ). If published, this will include your full peer review and any attached files.

**Do you want your identity to be public for this peer review?** For information about this choice, including consent withdrawal, please see our Privacy Policy .

Reviewer #2: **Yes: ** Marie Legein

Reviewer #3: No

Reviewer #4: **Yes: ** Arshad Iqbal

Reviewer #5: No

---

## [Author Response · Author response to Decision Letter 0]

20 Feb 2025

Dear Editor and Reviewers,

Thank you very much for your comments and professional advice. These opinions help to improve academic rigor of our article. Based on your suggestion and request we have made corrected modifications on the revised manuscript. We hope that our work can be improved again. We would like to show the details as File “Response to Reviewers”.

---

## [Decision Letter · Decision Letter 1]

5 Mar 2025

PONE-D-24-56644R1Effect of PGPR on growth and nutrient utilization of Elymus nutans Griseb at different temperaturesPLOS ONE

Dear Dr. wang,

Thank you for submitting your manuscript to PLOS ONE. After careful consideration, we feel that it has merit but does not fully meet PLOS ONE’s publication criteria as it currently stands. Therefore, we invite you to submit a revised version of the manuscript that addresses the points raised during the review process.

We look forward to receiving your revised manuscript.

Kind regards,

Massimiliano Cardinale, PhD

Academic Editor

PLOS ONE

Reviewers' comments:

Reviewer's Responses to Questions

**Comments to the Author**

1. If the authors have adequately addressed your comments raised in a previous round of review and you feel that this manuscript is now acceptable for publication, you may indicate that here to bypass the “Comments to the Author” section, enter your conflict of interest statement in the “Confidential to Editor” section, and submit your "Accept" recommendation.

Reviewer #2: (No Response)

Reviewer #3: (No Response)

Reviewer #4: All comments have been addressed

Reviewer #5: All comments have been addressed

2. Is the manuscript technically sound, and do the data support the conclusions?

Reviewer #2: Partly

Reviewer #3: Partly

Reviewer #4: Yes

Reviewer #5: Partly

3. Has the statistical analysis been performed appropriately and rigorously? 

Reviewer #2: I Don't Know

Reviewer #3: Yes

Reviewer #4: Yes

Reviewer #5: Yes

4. Have the authors made all data underlying the findings in their manuscript fully available?

Reviewer #2: Yes

Reviewer #3: Yes

Reviewer #4: Yes

Reviewer #5: No

5. Is the manuscript presented in an intelligible fashion and written in standard English?

Reviewer #2: Yes

Reviewer #3: Yes

Reviewer #4: Yes

Reviewer #5: Yes

6. Review Comments to the Author

Reviewer #2: Thank you for revising your manuscript thoroughly. Some comments remain, most importantly a comment raised by another reviewer concerning biological and technical replicates:

- In response to this comment, you have made adjustments in section 2.1, regarding the isolation of bacterial strains. However, it is in the experimental part of the manuscript i.e. in the incubation experiment (2.5) that biological and technical replicates are needed. Can you provide more detail here, i.e. how many replicates per condition, how many seeds per pot, how many measurements per parameter, did you repeat it at different timepoints, with freshly made microbial mixtures? Secondly, the plants were incubated in three incubators, were the conditions randomized between these incubators?

Other comments are minor:

- Abstract: The study started at an earlier stage then the demonstration of growth promoting effects. I suggest including more context: i.e. rhizosphere bacteria were isolated (at this stage it is still unknown if these are PGPR), selected based on known PGPR traits which were assessed in vitro, and finally tested in plants. This context underlines the novelty of the research: novel PGPR were selected specifically for Eng on Tibetan plateau.

- M&M: 2.6.3: briefly explain the methods used to determine these plant nutrient use efficiency, instead of including a reference.

- M&M: Change “excellent” strains to e.g. “selected” strains, to remain objective

- Results: Table 1: change in caption > 5 strains. Please include identification of the strains, preferably also in abstract and discussion. Are these species known for their PGPR effects?

- readability of the results and discussion remains difficult. Below a few recommendations to improve:

o describe the different treatments instead of uniquely referring to T1, T2 etc

o guide the reader to better understand figure 7; i.e. how was this figure made, what does it show, and how should it be interpreted?

o explain the differences between physiological efficiency (PE) and apparent recovery efficiency (RE) and how this is relevant for this research.

- Discussion: This phrase is misleading and should be nuanced: “Consistent with these findings, the culture substrate containing PGPR elicited IAA production (Table 1)” > The PGPR produced IAA on certain media, but PGPR eliciting IAA production in plants was not studied. Secondly, it is not sure that IAA was also produced in the conditions of the incubation experiment.

Reviewer #3: The title of this paper was different from now. So, some additional data and corrections are necessary to accept this manuscript as full length paper (comments are given as attached file). Hope, the author will response accordingly.

Reviewer #4: (No Response)

Reviewer #5: Authors have addressed my comments. The manuscript can now be accepted. I have not looked into comments from other reviewers.

7. PLOS authors have the option to publish the peer review history of their article (what does this mean? ). If published, this will include your full peer review and any attached files.

**Do you want your identity to be public for this peer review?** For information about this choice, including consent withdrawal, please see our Privacy Policy .

Reviewer #2: **Yes: ** Marie Legein

Reviewer #3: No

Reviewer #4: No

Reviewer #5: No

---

## [Author Response · Author response to Decision Letter 1]

13 Mar 2025

Response to Reviewers

Manuscript Number: PONE-D-24-56644 

Effect of PGPR on growth and nutrient utilization of Elymus nutans Griseb at different temperatures

Dear Editor and Reviewers,

Thank you very much for your comments and professional advice. These opinions help to improve academic rigor of our article. Based on your suggestion and request we have made corrected modifications on the revised manuscript. We hope that our work can be improved again. We would like to show the details as follows:

Reviewer #2:

1. Thank you for revising your manuscript thoroughly. Some comments remain, most importantly a comment raised by another reviewer concerning biological and technical replicates: In response to this comment, you have made adjustments in section 2.1, regarding the isolation of bacterial strains. However, it is in the experimental part of the manuscript i.e. in the incubation experiment (2.5) that biological and technical replicates are needed. Can you provide more detail here, i.e. how many replicates per condition, how many seeds per pot, how many measurements per parameter, did you repeat it at different timepoints, with freshly made microbial mixtures? Secondly, the plants were incubated in three incubators, were the conditions randomized between these incubators?

Answer: We are extremely grateful for the pertinent suggestions from the reviewers. In the incubation experiment (2.5), we carried out both biological and technical replications. For different treatments, we set up 3 replicates. Each pot contained 35 seeds, and each parameter was measured three times for each treatment. (Figure 2). Regarding the use of the mixed bacterial solution, we added it under the same time condition to minimize errors. Secondly, the plants were incubated in three incubators under the conditions of a light (16 h)/dark (8 h) cycle (with a light intensity of 350 μEm⁻² s⁻¹) and 60% humidity. Of course, the configuration and power of the incubators are also the same, and the three incubators are mainly used for three temperature culture experiments. The incubators have automatic temperature, humidity and light control devices, which can precisely control the temperature, humidity and light. In order to ensure that the plants receive the same light in the incubator during the cultivation process, as well as the same temperature and humidity conditions, the position of the samples in the incubator is randomly changed again every 2 days.

2. The study started at an earlier stage then the demonstration of growth promoting effects. I suggest including more context: i.e. rhizosphere bacteria were isolated (at this stage it is still unknown if these are PGPR), selected based on known PGPR traits which were assessed in vitro, and finally tested in plants. This context underlines the novelty of the research: novel PGPR were selected specifically for Eng on Tibetan plateau.

Answer: Thank you very much for the valuable comments provided by the reviewers. We have added more background information in the abstract part: rhizosphere bacteria were isolated and selected based on the known traits of PGPR. These traits were evaluated in vitro and finally tested in plants. Moreover, this background highlights that the PGPR of this study were specifically selected for EnG in the Qinghai-Tibet Plateau.

3. 2.6.3: briefly explain the methods used to determine these plant nutrient use efficiency, instead of including a reference.

Answer: Thank you for the reviewers' comments. We have briefly described the methods used to determine the nutrient use efficiency of these plants in Section 2.6.3.

4. Change “excellent” strains to e.g. “selected” strains, to remain objective.

Answer: Thank you for your suggestion. We have already changed the description of "excellent" strains to "selected" strains in the article.

5. Table 1: change in caption > 5 strains. Please include identification of the strains, preferably also in abstract and discussion. Are these species known for their PGPR effects?

Answer: Thank you very much for the reviewers' suggestions. We have already included the identification results of the strains in Table 1. Meanwhile, we have supplemented the background conditions for the isolation, screening and identification of the strains in the abstract.

6. readability of the results and discussion remains difficult. Below a few recommendations to improve:

describe the different treatments instead of uniquely referring to T1, T2 etc.

Answer: Thank you for your suggestion. We have modified the descriptions regarding the treatment methods in the Results and Discussion section, instead of referring to T1, T2, etc. individually.

7. guide the reader to better understand figure 7; i.e. how was this figure made, what does it show, and how should it be interpreted? explain the differences between physiological efficiency (PE) and apparent recovery efficiency (RE) and how this is relevant for this research.

Answer: Thank you for the insightful comments. Below is the elaboration:

Figure 7 visually integrates Mantel’s test and Pearson’s correlation analysis to illustrate relationships between plant physiological parameters and nutrient use efficiency indices. Appropriate additions and supplements have been made for the better understanding of the reader.

The physiological efficiency (PE) of N (NPE), P (PPE), and K (KPE) was calculated as the kg plant yield per kg N, P, and K uptake. The agronomic efficiencies (AE) of N (NAE), P (PAE), and K (KAE) were calculated as the yield increase per kg N, P, and K applied. The apparent recovery efficiency (RE) of N (NRE), P (PRE), and K (KRE) was calculated as the plant N, P, and K uptake (kg ha-1) per kg of N, P, and K applied. PE: Highlights PGPR’s role in optimizing nutrient assimilation under temperature stress (e.g., enhanced Chl synthesis via Fe mobilization). RE: Demonstrates PGPR’s ability to reduce fertilizer dependency by improving nutrient recovery (e.g., T3 vs. T4 treatments in Table 1). PE and RE collectively reveal PGPR’s dual impact on nutrient use strategies: PE reflects metabolic optimization, while RE quantifies practical fertilizer savings. For the better understanding of the readers, we have also added and supplemented the corresponding concepts and references in the methods section as appropriate.

8. Discussion: This phrase is misleading and should be nuanced: “Consistent with these findings, the culture substrate containing PGPR elicited IAA production (Table 1)” > The PGPR produced IAA on certain media, but PGPR eliciting IAA production in plants was not studied. Secondly, it is not sure that IAA was also produced in the conditions of the incubation experiment.

Answer: We are very grateful to you, the reviewer, for your valuable comments. We have made revisions to this discussion section.

Reviewer #3:

1. It is essential to add phytogenic tree of those 5 strains of PGPR for finding the geographical variation, if any.

Answer: Thank you for the reviewers' comments. I am very sorry that we did not conduct a study of phytogenic tree of those 5 strains of PGPR for finding the geographical variation. In this study, the sampling area is located in Nagqu Town, Seni District, Nagqu City, Tibet Autonomous Region, with the geographical coordinates of 31°26′N latitude and 92°07′E longitude, and an average elevation of 4,450 meters. Three sampling sites with well-grown and disease-free Elymus nutans Griseb (EnG) were randomly selected. The rhizosphere soil of Elymus nutans Griseb was collected by the root shaking method, and five mixed replicate samples were randomly collected at each sampling site (Figure 1). In conclusion, the strains were collected from natural grasslands on the northern Tibetan Plateau with little geographic differentiation.

2. In Fig 3c: the amounts of ‘Chl’ in both T1 and T3 at 10 &150C were obtained almost same. Again, Phosphorus is a key element that affects chlorophyll levels in plants. Here, it was not discussed- why ‘Chl’ amounts were obtained similar rather NPK was used in T3?

Answer: Thank you for the reviewers' comments. In this study, the amounts of “Chl” in T1 and T3 were almost the same at 10 °C and 15 °C. The possible reason is that the PGPR bacterial liquid was added in both T1 and T3 treatments. PGPR can promote the growth and development of plant roots by secreting plant hormones such as IAA, thus improving the nutrient absorption efficiency of plants. In the T1 treatment, although nitrogen, phosphorus and potassium fertilizers were not added, PGPR might decompose the insoluble nutrients in the culture medium through its growth-promoting characteristics, enabling plants to utilize the nutrients in the culture medium, including phosphorus, more effectively and thus maintain a relatively high chlorophyll content. In response to this, we have also added the relevant discussions of this part in the article.

3. It is also needed to put the Siderophores test results that represent iron (Fe) accumulation. Iron is involved in the manufacturing process of Chlorophyll (Chl); helps plants growth and development via Photosynthesis and in many metabolic processes.

Answer: We extend our heartfelt gratitude for your meticulous review of our article and the insightful suggestions you have offered. Concerning your recommendation to include the test results of siderophores associated with iron (Fe) accumulation, we have given this proposal serious consideration and are now providing the following clarifications.

Our study aimed to investigate the growth-promoting and resistance mechanisms exhibited by plant-growth-promoting rhizobacteria (PGPR) in Elymus nutans Griseb under various temperatures. Our primary focus was on the impact of PGPR on the growth and nutrient uptake efficiency of Elymus nutans Griseb, as well as its responsiveness to differing temperature conditions. While iron is vital for plant development, particularly in the synthesis of chlorophyll, photosynthesis, and metabolic processes, it was not the primary focus of our investigation.

We acknowledge the significance of iron accumulation and the pivotal role of siderophores in the interactions between plants and microorganisms. We further recognize the considerable potential for further exploration in this research area. Through active brainstorming, we are currently focusing on the relationship between iron (Fe) and photosynthesis in our ongoing field application experiments of mycorrhizal fertilizers, with the aim of obtaining valuable data. We intend to conduct more in-depth research in future studies, focusing specifically on the relationship between iron and chlorophyll, as well as the influence of PGPR on the iron uptake and utilization by plants. This comprehensive investigation will not only enhance our understanding of the growth-promoting mechanisms of PGPR but also contribute valuable data to the field.

We reiterate our appreciation for your valuable feedback. We trust that you appreciate our current circumstances. We are committed to actively pursuing this crucial topic in our future research endeavors and look forward to contributing more significant findings to the advancement of this field.

4. The shelf life and population of PGPR was not evaluated in that substrate. To validate this finding and farmers use, such data is needed.

Answer: First of all, thank you very much for your valuable suggestion, it is indeed very important to supplement the data on shelf life and colony counts of PGPR in substrates to validate the research findings and practical applications. Of course, we still apologize to you that we neglected to record the shelf life and colony number cycle of PGPR due to the time cycle and subsequent development and utilization of the bacterial fertilizer, because we have been carrying out the improvement and configuration of the bacterial fertilizer, so our strains have been in the state of regular activation. However, in order to ensure the rigor and scientificity of this potting experiment, we took out the samples kept in -80℃ for determining the microbial metabolism of the substrate and verified the activation, and the specific results are attached in the following figure, because considering the structure of the overall article, we hope that we can be allowed to use this result only for answering your questions, after all, it is the shelf-life of the PGPR which has not been systematically and completely recorded. By our activation and counting assay, based on the substrates at the end of the potting experiment (substrates kept in an ultra-low temperature refrigerator), the number of colonies in the substrates with added bacterial solution was significantly higher than in the control. Thank you for your valuable comments, and of course we hope to document its activity and shelf-life in bacterial fertilizers in a complete manner during the subsequent development of the bacterial fertilizer.

---

## [Decision Letter · Decision Letter 2]

30 Mar 2025

PONE-D-24-56644R2Effect of PGPR on growth and nutrient utilization of Elymus nutans Griseb at different temperaturesPLOS ONE

Dear Dr. wang,

Thank you for submitting your manuscript to PLOS ONE. After careful consideration, we feel that it has merit but does not fully meet PLOS ONE’s publication criteria as it currently stands. Therefore, we invite you to submit a revised version of the manuscript that addresses the points raised during the review process.

We look forward to receiving your revised manuscript.

Kind regards,

Massimiliano Cardinale, PhD

Academic Editor

PLOS ONE

Journal Requirements:

Additional Editor Comments (if provided):

Academic Editor:

Both reviewers accepted the manuscript but pointed out that the abstract is neither fluent nor clear. Therefore, I read it carefully and I found that the language is below the minimal scientific standard. For example:

- "In order to show the growth-promoting effects of selected PGPR isolated from the rhizosphere soil of Elymus nutans Griseb on the Tibetan Plateau." cannot be a stand-alone sentence: main clause is missed;

- "By isolating and purifying the rhizosphere bacteria from the rhizosphere soil of Elymus nutans Griseb (EnG) in the Qinghai-Tibet Plateau, use the selective culture medium to determine whether the strains have plant growth-promoting ability and measure the magnitude of their plant growth-promoting ability". This sound as a protocol where someone is giving you instructions.

I read the rest of the manuscript and I found further parts where the language is poor. Just as a few examples:

- "Five selected strains (S1, S2, S3, S4 and S5) with high growth promoting ability were screened by the determination of growth-promoting ability". Really confusing and repetitive.

- "Inoculate the purified strains onto Nitrogen free medium (NFM)..." This sound as a protocol where someone is instructing you to inoculate the strain on different media. In the article, you must write "Strains were inculated onto NFM..." etc.

Therefore, it is necessary that the manuscript is revised by a native english speaker before it can be accepted and published.

Another important aspect to be clarified concerns the statistical analysis: it is stated that both least significant difference (LSD) and Tukey test (HSD) were used. However, in the graphs where post-hoc letters are reported, it is not indicated which post-hoc method was used. Considering that these two post-hoc tests are very different (LSD the most permissive, HSD the most stringent one), it is really necessary that this is clearly indicated: significant differences assessed by HSD have a much greater value that those assessed by LSD!!

Reviewers' comments:

Reviewer's Responses to Questions

**Comments to the Author**

1. If the authors have adequately addressed your comments raised in a previous round of review and you feel that this manuscript is now acceptable for publication, you may indicate that here to bypass the “Comments to the Author” section, enter your conflict of interest statement in the “Confidential to Editor” section, and submit your "Accept" recommendation.

Reviewer #2: All comments have been addressed

Reviewer #3: All comments have been addressed

2. Is the manuscript technically sound, and do the data support the conclusions?

Reviewer #2: Yes

Reviewer #3: Yes

3. Has the statistical analysis been performed appropriately and rigorously? 

Reviewer #2: No

Reviewer #3: Yes

4. Have the authors made all data underlying the findings in their manuscript fully available?

Reviewer #2: Yes

Reviewer #3: Yes

5. Is the manuscript presented in an intelligible fashion and written in standard English?

Reviewer #2: Yes

Reviewer #3: Yes

6. Review Comments to the Author

Reviewer #2: Authors have addressed my comments. The added text in the abstract could be written more fluently, but the manuscript can be accepted.

Reviewer #3: See the attached file. In abstract, there is also a comment. It is suggested to solve this query and correct the word.

hope you will formulate this PGPR in future for commercial use.

7. PLOS authors have the option to publish the peer review history of their article (what does this mean? ). If published, this will include your full peer review and any attached files.

**Do you want your identity to be public for this peer review?** For information about this choice, including consent withdrawal, please see our Privacy Policy .

Reviewer #2: **Yes: ** Marie Legein

Reviewer #3: **Yes: ** Prof. Dr. Mohammad Delwar Hossain

---

## [Author Response · Author response to Decision Letter 2]

10 Apr 2025

Dear Editor and Reviewers,

Thank you very much for your comments and professional advice. These opinions help to improve academic rigor of our article. Based on your suggestion and request we have made corrected modifications on the revised manuscript. We hope that our work can be improved again. We would like to show the details as "Response to Reviewers".

---

## [Editor Report · Decision Letter 3]

11 Apr 2025

Effect of PGPR on growth and nutrient utilization of Elymus nutans Griseb at different temperatures

PONE-D-24-56644R3

Dear Dr. wang,

We’re pleased to inform you that your manuscript has been judged scientifically suitable for publication and will be formally accepted for publication once it meets all outstanding technical requirements.

Kind regards,

Massimiliano Cardinale, PhD

Academic Editor

PLOS ONE

Additional Editor Comments (optional):

Thank you for revising the manuscript according to my concerns.

Please, at the proofs correction step, remember to correct one sentence in the paragraph 2.3. "Identification of PGPR strains" as follows: "The 16S rRNA gene was amplified by polymerase chain reaction (PCR) ... ..."
---

## [Editor Report · Acceptance letter]

PONE-D-24-56644R3

PLOS ONE

Dear Dr. Wang,

I'm pleased to inform you that your manuscript has been deemed suitable for publication in PLOS ONE. Congratulations! Your manuscript is now being handed over to our production team.

Kind regards,

on behalf of

Dr. Massimiliano Cardinale

Academic Editor

PLOS ONE